# Assessment of geodetic velocities using GPS campaign measurements over long baseline lengths

Huseyin Duman[1] and Dogan Ugur Sanli[1]

[1]Department of Geomatic Engineering, Yildiz Technical University, Istanbul, 34220, Turkey

**Correspondence:** Dogan Ugur Sanli (usanli@yildiz.edu.tr)

**Abstract.** GPS campaign measurements are frequently used in order to determine geophysical phenomena such as tectonic motion, fault zones, landslides, and volcanoes. When observation duration is shorter, the accuracy of coordinates are degraded and also the accuracy of point velocities are affected. The accuracies of the geodetic site velocities from a global network of the International GNSS Service (IGS) stations were previously investigated using only PPP. In this study, we extend that study in which site velocities will also be assessed including fundamental relative positioning. PPP derived results will also be evaluated to see the effect of JPL reprocessed products, single receiver ambiguity resolution, repeating survey campaigns minimum 3 days at the site, and eliminating noisier solution prior to the year 2000. Globally distributed 18 continuously operating IGS stations were chosen to create synthetic GPS campaigns. GPS data were processed comparatively using GAMIT/GLOBK v10.6 and GIPSY/OASIS II v6.3. The data of synthetic campaign GPS time series were processed using a regression model accounting for the linear and seasonal variation of the ground motion. Once accepting the velocities derived from 24 hour sessions as the truth, the results from sub-sessions were compared with the results of 24 hours and hypothesis testing was applied for the significance of the differences. The major outcome of this study is that at global scales (i.e. over long distances) with short observation sessions, the fundamental relative positioning produces results similar to PPP. The reliability of the velocity estimation for GPS horizontal baseline components has now been improved to about 85% on the average for observation durations of 12 h.

## 1 Introduction

GPS measurements were gathered from campaign surveys from the end of 80s through mid 90s. As of the emerge of continuous GPS in the early 90s and after the release of IGS official orbit in 1994 campaign GPS measurements were combined with continuous GPS. By doing this, researchers wanted to take the advantage of episodic GPS measurements accumulated over the past 10 years. Campaign GPS measurements were mainly referred to monitor the global sea level in an attempt to decouple crustal motion from the actual sea level rise (Bingley et. al., 2001) and to monitor tectonic motion (Zhang et. al., 1997; Dixon et. al., 2000; Reilinger et.al., 1997).

Zhang et. al. (1997) studied stochastic properties of continuous GPS (19-month long) data from permanent stations. They then extended their time series by extrapolation to 5 years and generated campaign GPS measurements. Velocities were estimated from those synthetically generated campaign measurements. Velocity estimation (i.e. standard) errors of campaign

measurements were assessed employing white noise and colored noise models. The stochastic model derived from continuous measurements was recommended for finding the standard errors of deformation rates obtained from campaign measurements.

Dixon et. al. (2000) used both campaign and continuous GPS to interpret the motion of Sierra Nevada Block. They followed a similar procedure as given in Zhang et. al. (1997) and determined the stochastic model of their continuous measurements to later calibrate the velocity error of their combined time series. Bingley et. al. (2001) followed a similar procedure in finding crustal motion in monitoring of the sea level, and velocities of campaign GPS measurements were computed in the combination model using the suggestions from Zhang et. al. (1997) and Mao et. al. (1999).

In the new milenium, many studies (Vernant et. al., 2004; Serpelloni et. al., 2007; Hollenstein et. al., 2008; Chousianitis et. al., 2015; Bitharis et. al., 2016) in which GPS velocity fields have been used to facilitate tectonic and geodynamic research were performed applying the procedure detailed in Zhang et. al. (1997), Mao et. al. (1999), and Dixon et. al. (2000). However, many others that employ campaign measurements were performed with the procedure; 1-day per year, collecting GPS measurements with only 8-10 hour observation session during the measurement day (Miranda et. al., 2012; Elliott et. al., 2010; Rontogianni, 2010; Ashurkov et. al., 2011; Ozener et. al., 2013; Tran et. al., 2013; Catalão et. al., 2011). Unluckily, the velocities from such campaigns vere estimated from only a couple of years (i.e. using only 2-3 estimates). Ambiguity resolution from the GPS baseline processing and hence the positioning accuracy as well as velocity estimation from the above campaigns were deteriorated due to the fact that GPS baseline solutions were produced for long baselines up to 2000 km with only 8-10 h of the data.

To critisize the above studies Akarsu et. al. (2015) designed a global IGS study in which station velocities from 8-12 h GPS campaigns were assessed against those of the 24 h GPS campaigns. Differing from the studies which aim to assess the "standard error level of estimated velocities" from campaign measurements (i.e. Zhang et. al. (1997); Dixon et. al. (2000); Mao et. al. (1999)),Akarsu et. al. (2015) emphasized the term "accuracy of velocities" by which the accuracy of velocities estimated from 8-12 h observation time series is assessed against those of 24 h observations which are taken as the truth. They used the PPP online module of GIPSY/OASIS II (APPS) to analyze the GPS data. The results revealed that only 30-40% of the horizontal and none of the vertical velocities were comparable to the accuracy derived from 24 h campaigns.

On the other hand, the analysis of Akarsu et. al. (2015) at the time did not include some of the improvements due to recent developments in regard to the GIPSY/OASIS II processing, such as new JPL reprocessed products (i.e. orbits and clocks) and single receiver ambiguity solution. Considering those developments and adding some extra measures to the surveying procedure, we believe the success rate of estimated velocities will be improved. The extra measures mentioned above consider carrying out campaign GPS measurements in 3 consecutive days with overlapping sessions and including GPS days with ionospheric kappa index less than 4. Furthermore, the PPP results produced will be assessed with fundamental relative positioning using GAMIT/GLOBK analysis with the hypothesis "GPS relative positioning over long baseline lengths with short occupation durations should produce positioning information equivalent to PPP campaign results". To handle this experiment a global network of 18 IGS stations were selected and the GPS data were analyzed using GAMIT and GIPSY. Synthetic GPS campaigns were created from the continuous observations with 8, 12 and 24 h sessions. GPS data were processed for all sessions to form north, east and up campaign time series. Velocities derived from all three GPS components calculated from both 8 and 12 h

sub-sessions were compared with the velocities from 24 h which were accepted as the truth. The differences from the truth were statistically tested and the result were interpreted.

## 2 Methodology

### 2.1 GPS Data Analysis

GPS data were downloaded in Receiver Independent Exchange (RINEX) format with 30 sec. intervals from the Scripps Orbit and Permanent Array Center (SOPAC) which is one of the data archives of the International GNSS Service (IGS) at http://sopac.ucsd.edu/. The IGS stations used in the study are demonstrated in Figure 1. First of all, to determine the horizontal velocities for each of the stations we selected three successive days in October each year for the years 2000 through 2015. Akarsu et. al. (2015) did the similar sampling using only 1 day in a year. This is the procedure followed by many of the GPS

experiments using repeated surveys (Aktuğ et. al., 2009, 2013; Dogan et. al., 2014; Koulali et. al., 2015; McClusky et. al., 2000; Ozener et. al., 2010; Tatar et. al., 2012). Using 3 consecutive days here we believe we increased the reliability of the solutions. A treatment in regard to the solar activity, which was missing in Akarsu et. al. (2015), was also taken into consideration (i.e. days with kappa index $\leq 4$) here. In addition, three successive days in every month were included for the processing of the vertical component. In order to model the significant annual signal on GPS heights, here we did the sampling monthly. The

GPS data were segmented into sub-sessions as listed in Table 1 in order to generate the repeated GPS measurements.

### 2.1.1 GAMIT/GLOBK Processing

The GPS data were processed with GAMIT/GLOBK v10.6 software for relative point positioning (Herring et. al., 2006a, b) and with GIPSY/OASIS II v6.3 for PPP (Zumberge et. al., 1997). The elevation cut-off angle was set to 7 degrees on both software.

The processing of the GPS data using GAMIT/GLOBK was conducted in three steps (Feigl et. al., 1993; McClusky et. al., 2000; Reilinger et. al., 2006; Tatar et. al., 2012; Dong et. al., 1998; Cetin et. al., 2018). We selected 18 globally scattered IGS stations. At first, the loosely constrained station coordinates, atmospheric zenith delays of each points, and Earth Orientation Parameters (EOP) were estimated using doubly differenced GPS phase measurements and IGS final products. Global Pressure and Temperature 2 (GPT2) mapping function developed by Lagler et. al. (2013) was used to model the delay in the atmosphere.

The ocean tide loading correction was applied using FES2004 model of Lyard et. al. (2006). Ambiguities were on the average resolved with 90% success for the wide-lane and 80% success for the narrow-lane (Figure 2).

Secondly, GLOBK was used to estimate the point coordinates and velocities from a combined solution comprising the daily loosely constrained estimates, EOP values, orbit data, and their covariance through Kalman Filtering. We used the IERS (International Earth Rotation and Reference Systems Service) Bulletin B values for Earth rotation parameters. Since our study

was initially designed to be a global experiment, in this step we did not extra enlarge our network with more globally scattered and loosely constrained IGS stations.

In the last step, the reference frame was realized on each day through "generalized constraints". Iterations were applied on the initially chosen 18 IGS stations, and about 5 bad sites were eliminated after 4 iterations. The reference frame was realized on each day employing a reliable set of round 13 IGS stations in the ITRF2008 no-net-rotation (NNR) frame (Altamimi et. al., 2012). The reliability of the IGS stations was characterized with GPS days which do not contain the effect of bad ionospheric conditions with kappa index values smaller than 4, having at least 95% data coverage, being available on the common days, and repeating in 3 consecutive days with overlapping sessions.

The processing strategies described above were applied to each subset of sessions listed in Table 1. The coordinate values for all sub-sessions were transformed to the topocentric system consisting of East, North, and Up. The time series of the site ZIMM from relative positioning and PPP solutions for all sub-sessions were illustrated in Figure 3.

## 2.1.2 GIPSY-OASIS II Processing

We used JPL final precise (flinnR) orbits and clocks in the analysis. The (precise point positioning) PPP module of GIPSY/OASIS II v6.3 was developed by Zumberge et. al. (1997). In GIPSY analysis, final orbits and clocks are determined from a global network solution. The results were represented using the International Earth Rotation Service's reference system ITRS (Petit and Luzum, 2010), as realized through the reference frame ITRF2008 (Altamimi et. al., 2012). Tropospheric Zenith Wet Delay was modelled as a random-walk parameter with a variance rate of 5 mm$^2$ per hour and wet delay gradient with a variance rate of 0.5 mm$^2$ per hour. The dry troposphere was modelled using GPT2 mapping function a priori zenith conditions (Lagler et. al., 2013). Pseudo-range and carrier phase observations were employed to eliminate the ionospheric delay using L1 and L2 data combination. Kedar et. al. (2003) model was used to eliminate the effect of second order ionosphere. Satellite and receiver antenna phase centre variation (APV) maps were automatically applied following the IGS standards (Haines et. al., 2010). Desai (2002) model was used to eliminate the effect of ocean tide loading.

## 2.2 Velocity Estimation and Statistical Tests

In this section, with the motivation from Akarsu et. al. (2015), the estimation of an IGS site velocity and the related statistical tests will be explained. In Figure 3, the comparison of all sub-session coordinate time series for all three GPS baseline components from both software results have been shown. Once looking at the time series, all sub-sessions get along well with each other. For the horizontal components that are East and North, the variations are almost perfectly linear, and this shows us the tectonic motion clearly. To estimate the linear variation (i.e. the velocity) the model of

$$x_i = a.t_i + b + o_i.x_{off} + v_i \qquad (1)$$

was used. There $x_i$ represents any coordinate value, $a$ site velocity, $t_i$ the time, $b$ the intercept, and $v_i$ the residuals. In a GPS solution time series there are additional terms such as to clarify the sudden displacements due to earthquakes. Then, Eq. (1) is expanded to include an offset value $x_{off}$ and the corresponding coefficient $o_i$ (Montillet et. al., 2015). For instance in

our analysis, the stations AREQ and USUD include offset values in their time series due to earthquakes. For all stations, the velocity estimations were calculated using Eq. (1) by means of the least square estimation.

The vertical component additionally includes significant seasonal variation. The coordinate time series for vertical components contain repeating annual cycle stemming from hydrological and atmospheric loading (Blewitt and Lavallée, 2002). Santamaría-Gómez et. al. (2011) noticed seasonal motions in smaller periods like 3 and 4 months and diminishing amplitudes in GPS time series from continuously operating stations . Given these circumstances, it is not sufficient to determine vertical velocities with a linear model. The seasonal model we use here take into account the annual and semi-annual periodicities;

$$x_i = a.t_i + b + o_i.x_{off} + \sum_{n=1}^{q}[c_n.cos\frac{2\pi.t_i}{T_n} + d_n.sin\frac{2\pi.t_i}{T_n}] + v_i \tag{2}$$

where $q = 2$, $T_1 = 1$ year and $T_2 = 0.5$ year. The use of the offset parameter is the same as in the horizontal assessment. Furthermore, $R^2$ value known as "coefficient of determination" was computed in regression analysis as a statistical tool, which shows how well the data fit to the model estimated. For any coordinate component from a regression analysis, computation of $R^2$ is given with;

$$R^2 = 1 - \frac{\sum_{i=1}^{n} \hat{v}_i^2}{\sum_{i=1}^{n}(x_i - \overline{x})^2} = 1 - \frac{\sum_{i=1}^{n}(x_i - \hat{x}_i)^2}{\sum_{i=1}^{n}(x_i - \overline{x})^2} \tag{3}$$

where $\hat{v}_i = x_i - \hat{x}_i$ refers to the regression values $\hat{x}_i = \hat{a}t_i + \hat{b}$ based on the least square estimation ($\sum_{i=1}^{n} v_i^2 = min$), and $\overline{x}$ is the arithmetic average of $n$ number of measurements used in the estimation. The velocity estimation results and $R^2$ values from both processing strategies for the station of ZIMM have been listed in Table 2. Almost all $R^2$ values of the horizontal components in Table 2 for both software are at the level of 0.99 whereas those of the vertical ones range from 0.27 to 0.49. It is because the up component sampled using monthly data is not obviously linear. Parallel to the low $R^2$ values, the estimated velocities also fluctuate larger for the vertical component.

For all the stations in the IGS network, the solutions from sub-sessions were compared with the solutions (i.e. velocities) of 24 hours accepted as the truth. The statistical assessments of hypotheses were carried out in three steps. Then, the equivalancy between the unit variance derived from LSE of the sub-session given in Table 1 and with that of the 24h session was tested. The relevant hypothesis testing was set to be $H_0 : \sigma_{24}^2 = \sigma_s^2$ and $H_A : \sigma_{24}^2 \neq \sigma_s^2$ where $\sigma^2$ represents the unit variance and subscripts represent the observation session. A hypothesis testing based on F-distribution was applied to check the equivalancy of the variances. In the case unit variances were found to be equivalent, Student's t test was applied whether or not the velocity estimated from the campaign GPS significantly differs from the velocity derived from continuous GPS. The null hypothesis was set to be ($H_0 : a_{24} = a_s$) against the alternative hypothesis ($H_0 : a_{24} \neq a_s$) where $a_i$ denotes the velocity values in Eq. (1) and (2). Briefly, it was tested whether or not there is a significant difference between the results of 24 hour solutions and those of the sub-sessions. In these statistical tests, the degree of freedom values for the horizontal components where approximately 42 whereas the degree of freedom for the vertical component was about 345. The degree of freedom varied with respect to the number of insignificant parameters from the LSE.

## 3 Results and Discussion

As described in the previous section, the time series generated from all sessions of each continuous GPS site were analyzed. The coefficient of determination (i.e. $R^2$), which shows how well the data fits to the model, is computed according to Eq. (3). Tables 3, 4 and 5 compare the results of sub-sessions with those of the 24-hour statistically. Tables generally consist of two
columns, which includes the relative evaluation results from GAMIT / GLOBK v10.60 and the PPP results from GIPSY-OASIS II v6.3 software. In each column, $R^2$ values and hypothesis test results are given.

    Hypothesis test results are based on a 95% confidence level. If the hypothesis $H_0$ is accepted, it is shown that there is no statistically significant difference from the geodetic site velocities from sub-sessions to those from 24h session results. If the hypothesis $H_0$ is rejected (only expressed in **bold**), a posteriori unit variance obtained from the least square estimation is
statistically different from the 24 h one based on the F-test, that is, the models used for the geodetic velocity estimation are not equivalent. Furthermore, both the results expressed in **bold and underlined** indicate that the model is equivalent, but the deformation rate from 24-hour session is statistically different from the velocities from the sub-sessions based on the Student's t test.

    In Fig. 3, the subplots of the horizontal component clearly show the character of the tectonic motion linearly. In this context,
once Table 4 and 5 are examined, it is obviously seen that $R^2$ values estimated from all sessions are so close to 1 except for GUAM, KERG, USUD (in Table 4) and DAV1 (in Table 5). For instance, the motion in USUD is thought to be due to the postseismic relaxation.

    Success rates of velocity estimation from PPP and relative positioning are illustrated in Fig. 4. There blue bars are for 8h- and orange bars are for 12h-sessions. With the success rate here, we mean the success of velocity estimation from short sessions
when velocity estimation from 24h- is taken as the truth. Dashed pattern shows PPP results, whereas no pattern is for relative positioning. First of all, success rates for the horizontal components vary from 40 to 90%. Furthermore, the rates from 12 h sessions are higher than those of the 8 h sessions as expected. Note that the horizontal success rates from PPP are higher than those of the relative positioning, formed over long baselines to use in tectonic studies.

    The fact that the accuracy of the vertical component is worse than that of the horizontal component is often expressed in
the literature and in practice by many researchers. Therefore, the repeatabilities for this component are larger, and the seasonal effects are much more apparent than the horizontal ones. For this reason, the values of $R^2$ in Table 5 are much lower than those calculated from the horizontal component (around 0.40). Likewise, the results of the hypothesis test were rejected at a higher rate. Both for PPP and relative positioning, success rates for the vertical component are so low varying from about 5 to 15%. These rates are almost same for both methods.

Overall, the success rates of 12h- solutions are higher than those of 8h- solutions, the systematic effect acting on shorter sessions is varried and greater. For both positioning methods, the east component has greater success than the north one with regard to the truth. The success rates in Fig. 4 are higher for PPP than relative positioning because in the GAMIT/GLOBK processing are formed long baseline lengths. Over long baseline lengths, troposphere and ionosphere modelling become difficult, orbit errors accumulate, and hence ambiguity resolution becomes worse.

The reliability of velocity estimation from short GPS campaigns using PPP has been improved here when comparing results with those of Akarsu et. al. (2015). By the improvement we mean that the statistical agreement between the velocity estimated from short GPS observation and those of 24h sessions is higher here. The improvement in the horizontal component is 35% and 40% for 8h and 12h respectively. The vertical component was improved 4% and 17% for 8h and 12h respectively. This improvement might be ascribed to a few developments in the analysis procedures. First of all, here we used GPS time series spread over the years 2000 onwards. In other words, the noisier part 1990-2000 which might have affected the quality of estimated velocities in Akarsu et. al. (2015) is eliminated. Second, the analysis was performed with reprocessed orbits and clocks. JPL changed its orbit and clock estimation strategy as of the year 2007 (Hayal and Sanli, 2016). Third, GIPSY single receiver ambiguity resolution was further improved the accuracy of PPP (Bertiger et. al., 2010). Bertiger et. al. (2010) and Hayal and Sanli (2016) showed how positioning accuracy was improved with reprocessed JPL products and single receiver ambiguity resolution. Reprocessing especially improved the east component and this is correlated with the findings in this paper. Fourth, campaign measurements were performed in three consecutive days (i.e. the sampling was made such that IGS data were processed sellecting 3-consecutive days from the archive). Therefore, it was possible to eliminate the outlier solution from the processing. Finally, GPS campaigns were selected from the days in which the effect of geomagnetic storms is eliminated.

Eckl et. al. (2001) showed that using proper ambiguity resolution, troposphere modelling, and IGS precise orbits relative positioning performs uniformly, i.e. is not dependent on baseline length, for baseline lengths shorter than 300 km. In this experiment, GAMIT/GLOBK relative positioning used baseline lengths longer than 300 km. This degraded the accuracy of positioning and hence the velocity estimation of relative positioning. This was even achieved with slightly coarser accuracy than the PPP positioning. Based on relative positioning, BERNESE processing also gave similar results in Duman and Sanli (2016). Many studies in the literature monitoring tectonics with long baselines to stable plates need to take this into account (Ayhan et. al., 2002; Aktuğ et. al., 2015; Reilinger et. al., 2006; Ozener et. al., 2010).

## 4  Conclusions

We incorporated relative positioning in the determination of the accuracy of GPS site velocities from GPS campaigns (i.e. Observation sessions shorter than 24 h). Relative positioning results were produced from GAMIT/GLOBK. The results were also compared with PPP solutions derived from GIPSY/OASIS II. A global experiment for proper sampling was adopted using the IGS network.

The results indicate that relative positioning using long baseline lengths and short observation sessions produces similar results as with PPP. The accuracy is slightly coarser for horizontal positioning and slightly better for vertical positioning. Previously it has been noted that the accuracy of relative positioning does not depend on baseline length if baselines are shorter than 300 km. In the GAMIT/GLOBK processing here, reference points were chosen longer than 300 km.

It has also been noted that the accuracy of GPS site velocities derived from short observation sessions using PPP was improved here compared to previous studies. This is ascribed to the fact that the new GIPSY PPP runs with a new ambiguity resolution algorithm called single receiver ambiguity resolution. This especially improved the east component and the east

results in this study show better accuracy. Furthermore, the analysis was performed using reprocessed JPL orbits and clocks. The contribution of JPL reprocessed products to positioning was previously discussed among researchers. Differing from the previous studies, our sampling here also comprises GPS days freed from the effect of geomagnetic storms. In addition, repeating campaign GPS measurements in three consecutive days helps removal of a bad solution from the analysis. The noisier IGS time series between 1990 and 2000 was not used. If the user takes into account the above listed factors in the planning of their field works they should expect similar types of accuracy levels.

In this study, the horizontal velocity accuracy of GPS campaigns with 12 h observation sessions from PPP seems to reach the confidence level of about 85%. However, the reliability of vertical velocities is very poor and about at 20% level. This means that tectonic studies trying to use the daylight as the observation duration would still produce poorer accuracy than the expected 95%. Of course this result is based on GPS solutions only, one should expect 95% levels once solutions are compiled from multi-GNSS experiments. Although the accuracy of velocity estimation was improved about 40% for horizontal and 20% for vertical positioning, it is still not at the desired confidence level.

*Acknowledgements.* The authors are grateful to the IGS for the distribution of GNSS data and to SOPAC for perfect archiving service of the data. The figures were plotted using "The Generic Mapping Tools" and we express our gratitude to the open source software designers. We extend our gratitude to Fatih Poyraz for the valued discussions on GAMIT/GLOBK processing. We also would like to thank to the two anonymous reviewers for their constructive comments.

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

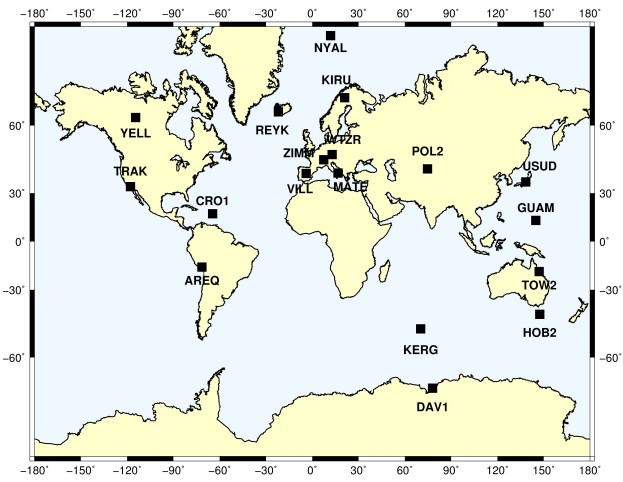

**Figure 1.** IGS continuous GPS sites used in the study.

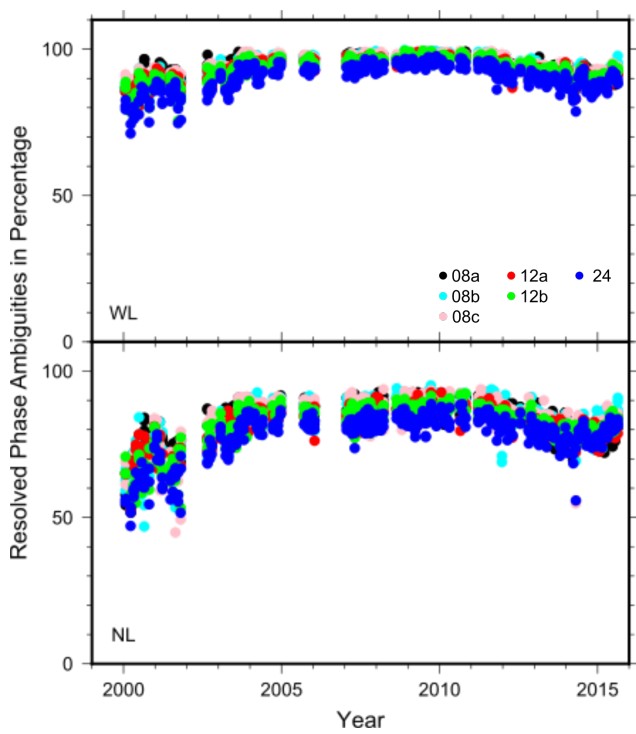

**Figure 2.** Daily fixed phase ambiguity resolution in percentage. WL: Wide-Lane and NL: Narrow-Lane ambiguity resolutions.

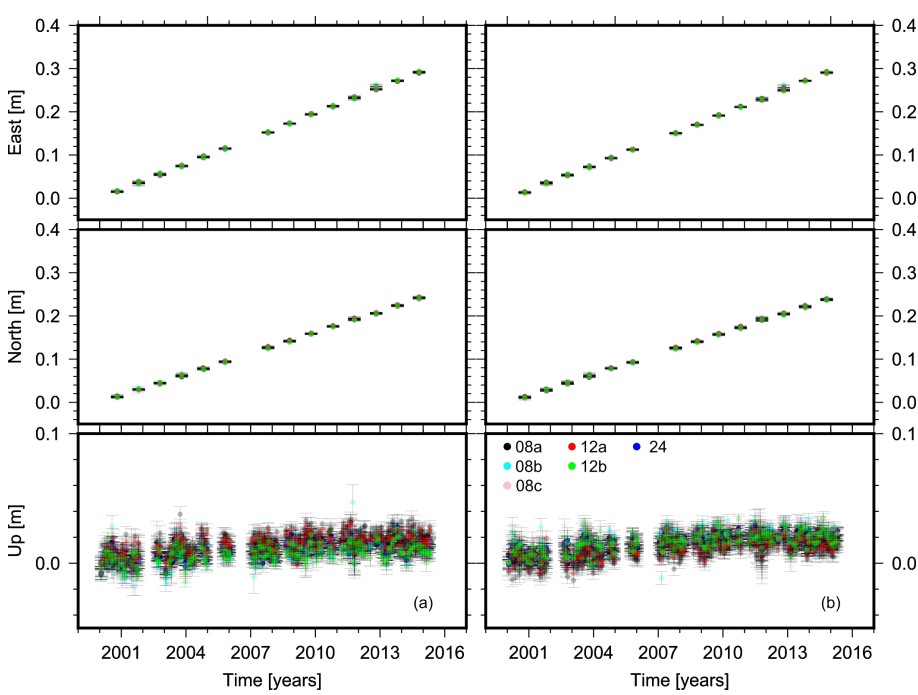

**Figure 3.** Time series of all sub-sessions for the site ZIMM from (a) relative positioning and (b) PPP solutions.

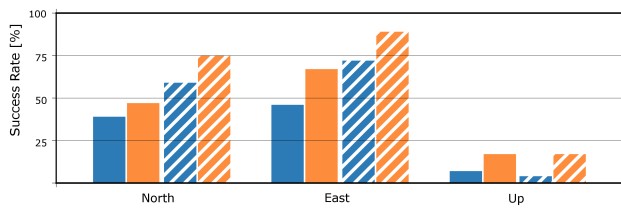

**Figure 4.** Success rates of 8h- and 12h-sessions. The velocity estimation from short sessions is compared with 24h results for each GPS component. Blue bars are for the 8h- and orange bars are for the 12h-sessions. Dashed patterns illustrate PPP estimates whereas no patterns are for the relative positioning.

**Table 1.** Segmented sub-sessions

| Session Length | Session Duration Times | | |
|---|---|---|---|
| [hours] | a | b | c |
| 8 | 00:00 - 08:00 | 08:00 - 16:00 | 16:00 - 24:00 |
| 12 | 00:00 - 12:00 | 12:00 - 24:00 | |

**Table 2.** Site velocities and $R^2$ values from both solutions for the site ZIMM

| GPS Baselines | | $R^2$ / Velocities [mm/yr] | | | | | |
|---|---|---|---|---|---|---|---|
| Observation Sessions | | 08a | 08b | 08c | 12a | 12b | 24 |
| GAMIT/GLOBK | East | 0.9998 | 0.9994 | 0.9998 | 0.9999 | 0.9995 | 0.9998 |
| | | 19.58 | 19.73 | 19.85 | 19.65 | 19.82 | 19.76 |
| | North | 0.9997 | 0.9991 | 0.9994 | 0.9997 | 0.9995 | 0.9997 |
| | | 16.25 | 16.21 | 16.29 | 16.34 | 16.21 | 16.28 |
| | Up | 0.4338 | 0.3095 | 0.2704 | 0.4099 | 0.3615 | 0.4714 |
| | | 1.06 | 1.06 | 0.67 | 0.96 | 0.76 | 0.91 |
| GIPSY/OASIS II | East | 0.9997 | 0.9991 | 0.9996 | 0.9997 | 0.9994 | 0.9997 |
| | | 19.69 | 19.77 | 19.78 | 19.73 | 19.77 | 19.75 |
| | North | 0.9996 | 0.9992 | 0.9989 | 0.9996 | 0.9991 | 0.9996 |
| | | 16.16 | 16.10 | 16.11 | 16.17 | 16.04 | 16.11 |
| | Up | 0.4233 | 0.3174 | 0.3433 | 0.4324 | 0.4005 | 0.4900 |
| | | 1.08 | 0.97 | 0.92 | 1.00 | 0.99 | 0.99 |

**Table 3.** For the north component, $R^2$ values and hypothesis test results for relative positioning and PPP

| Stations | GAMIT/GLOBK v10.60 | | | | | GIPSY/OASIS II v6.3 | | | | |
| --- | --- | --- | --- | --- | --- | --- | --- | --- | --- | --- |
| | 8 h | | 12 h | | 24 h | 8 h | | 12 h | | 24 h |
| | $R^2$ | Test Results | $R^2$ | Test Results | $R^2$ | $R^2$ | Test Results | $R^2$ | Test Results | $R^2$ |
| | 0.9816 | H0 accepted | 0.9810 | H0 accepted | 0.9811 | 0.9853 | H0 accepted | 0.9850 | H0 accepted | 0.9840 |
| AREQ | 0.9823 | H0 accepted | 0.9782 | H0 accepted | | 0.9839 | H0 accepted | 0.9833 | H0 accepted | |
| | 0.9792 | H0 accepted | | | | 0.9828 | H0 accepted | | | |
| | 0.9953 | **H0 rejected** | 0.9967 | H0 accepted | 0.9974 | 0.9980 | H0 accepted | 0.9980 | H0 accepted | 0.9984 |
| CRO1 | 0.9926 | **H0 rejected** | 0.9966 | H0 accepted | | 0.9970 | **H0 rejected** | 0.9968 | **H0 rejected** | |
| | 0.9960 | H0 accepted | | | | 0.9975 | H0 accepted | | | |
| | 0.9883 | **H0 rejected** | 0.9915 | **H0 rejected** | 0.9803 | 0.9817 | H0 accepted | 0.9847 | H0 accepted | 0.9825 |
| DAV1 | 0.9825 | **H0 rejected** | 0.9756 | H0 accepted | | 0.9804 | H0 accepted | 0.9744 | H0 accepted | |
| | 0.9720 | H0 accepted | | | | 0.9718 | H0 accepted | | | |
| | 0.7315 | H0 accepted | 0.7835 | H0 accepted | 0.7817 | 0.8922 | H0 accepted | 0.9037 | H0 accepted | 0.9020 |
| GUAM | 0.8178 | H0 accepted | 0.7880 | **H0 rejected** | | 0.9168 | H0 accepted | 0.8759 | H0 accepted | |
| | 0.7830 | H0 accepted | | | | 0.8702 | H0 accepted | | | |
| | 0.9998 | **H0 rejected** | 0.9999 | H0 accepted | 0.9999 | 0.9999 | **H0 rejected** | 0.9999 | **H0 rejected** | 0.9999 |
| HOB2 | 0.9998 | **H0 rejected** | 0.9998 | **H0 rejected** | | 0.9999 | H0 accepted | 0.9999 | H0 accepted | |
| | 0.9998 | **H0 rejected** | | | | 0.9999 | **H0 rejected** | | | |
| | 0.7352 | **H0 rejected** | 0.7891 | **H0 rejected** | 0.8809 | 0.8324 | H0 accepted | 0.8298 | H0 accepted | 0.8766 |
| KERG | 0.8470 | H0 accepted | 0.8152 | H0 accepted | | 0.8457 | H0 accepted | 0.8423 | **H0 rejected** | |
| | 0.8121 | **H0 rejected** | | | | 0.7685 | **H0 rejected** | | | |
| | 0.9992 | **H0 rejected** | 0.9994 | **H0 rejected** | 0.9993 | 0.9991 | H0 accepted | 0.9991 | H0 accepted | 0.9991 |
| KIRU | 0.9989 | **H0 rejected** | 0.9992 | H0 accepted | | 0.9985 | **H0 rejected** | 0.9987 | H0 accepted | |
| | 0.9992 | **H0 rejected** | | | | 0.9990 | **H0 rejected** | | | |
| | 0.9995 | **H0 rejected** | 0.9996 | **H0 rejected** | 0.9998 | 0.9994 | **H0 rejected** | 0.9995 | H0 accepted | 0.9997 |
| MATE | 0.9996 | **H0 rejected** | 0.9995 | **H0 rejected** | | 0.9992 | **H0 rejected** | 0.9994 | **H0 rejected** | |
| | 0.9993 | **H0 rejected** | | | | 0.9991 | **H0 rejected** | | | |
| | 0.9981 | **H0 rejected** | 0.9981 | **H0 rejected** | 0.9989 | 0.9992 | H0 accepted | 0.9992 | H0 accepted | 0.9993 |
| NYAL | 0.9979 | **H0 rejected** | 0.9988 | H0 accepted | | 0.9990 | H0 accepted | 0.9988 | **H0 rejected** | |
| | 0.9980 | **H0 rejected** | | | | 0.9988 | **H0 rejected** | | | |
| | 0.9797 | **H0 rejected** | 0.9755 | **H0 rejected** | 0.9841 | 0.9885 | **H0 rejected** | 0.9902 | H0 accepted | 0.9900 |
| POL2 | 0.9851 | H0 accepted | 0.9855 | **H0 rejected** | | 0.9855 | H0 accepted | 0.9879 | H0 accepted | |
| | 0.9832 | **H0 rejected** | | | | 0.9834 | **H0 rejected** | | | |
| | 0.9993 | H0 accepted | 0.9994 | **H0 rejected** | 0.9994 | 0.9992 | **H0 rejected** | 0.9994 | H0 accepted | 0.9996 |
| REYK | 0.9991 | H0 accepted | 0.9992 | H0 accepted | | 0.9995 | H0 accepted | 0.9995 | H0 accepted | |
| | 0.9991 | H0 accepted | | | | 0.9993 | H0 accepted | | | |
| | 0.9992 | **H0 rejected** | 0.9996 | H0 accepted | 0.9996 | 0.9998 | **H0 rejected** | 0.9998 | **H0 rejected** | 0.9999 |
| TOW2 | 0.9995 | **H0 rejected** | 0.9996 | **H0 rejected** | | 0.9999 | H0 accepted | 0.9999 | H0 accepted | |
| | 0.9996 | H0 accepted | | | | 0.9999 | **H0 rejected** | | | |
| | 0.9970 | H0 accepted | 0.9976 | H0 accepted | 0.9969 | 0.9982 | **H0 rejected** | 0.9961 | **H0 rejected** | 0.9993 |
| TRAK | 0.9931 | **H0 rejected** | 0.9926 | **H0 rejected** | | 0.9972 | **H0 rejected** | 0.9993 | H0 accepted | |
| | 0.9920 | **H0 rejected** | | | | 0.9986 | **H0 rejected** | | | |
| | 0.8624 | H0 accepted | 0.9008 | H0 accepted | 0.9162 | 0.9436 | H0 accepted | 0.9470 | H0 accepted | 0.9456 |
| USUD | 0.9182 | H0 accepted | 0.8921 | **H0 rejected** | | 0.9365 | H0 accepted | 0.9400 | H0 accepted | |
| | 0.8852 | H0 accepted | | | | 0.9402 | H0 accepted | | | |
| | 0.9990 | **H0 rejected** | 0.9992 | **H0 rejected** | 0.9995 | 0.9983 | H0 accepted | 0.9987 | H0 accepted | 0.9989 |
| VILL | 0.9992 | **H0 rejected** | 0.9997 | H0 accepted | | 0.9987 | H0 accepted | 0.9988 | H0 accepted | |
| | 0.9993 | H0 accepted | | | | 0.9983 | H0 accepted | | | |
| | 0.9995 | **H0 rejected** | 0.9996 | **H0 rejected** | 0.9998 | 0.9994 | H0 accepted | 0.9995 | H0 accepted | 0.9996 |
| WTZR | 0.9994 | **H0 rejected** | 0.9997 | **H0 rejected** | | 0.9985 | **H0 rejected** | 0.9995 | H0 accepted | |
| | 0.9997 | H0 accepted | | | | 0.9995 | H0 accepted | | | |
| | 0.9940 | **H0 rejected** | 0.9958 | H0 accepted | 0.9953 | 0.9970 | H0 accepted | 0.9973 | H0 accepted | 0.9970 |
| YELL | 0.9923 | **H0 rejected** | 0.9912 | **H0 rejected** | | 0.9961 | H0 accepted | 0.9937 | **H0 rejected** | |
| | 0.9879 | **H0 rejected** | | | | 0.9921 | **H0 rejected** | | | |
| | 0.9997 | H0 accepted | 0.9997 | H0 accepted | 0.9997 | 0.9996 | H0 accepted | 0.9996 | H0 accepted | 0.9996 |
| ZIMM | 0.9991 | **H0 rejected** | 0.9995 | **H0 rejected** | | 0.9992 | **H0 rejected** | 0.9991 | **H0 rejected** | |
| | 0.9994 | **H0 rejected** | | | | 0.9989 | **H0 rejected** | | | |

**Table 4.** For the east component, $R^2$ values and hypothesis test results for relative positioning and PPP.

| Stations | GAMIT/GLOBK v10.60 | | | | | GIPSY/OASIS II v6.3 | | | | |
|---|---|---|---|---|---|---|---|---|---|---|
| | 8 h | | 12 h | | 24 h | 8 h | | 12 h | | 24 h |
| | $R^2$ | Test Results | $R^2$ | Test Results | $R^2$ | $R^2$ | Test Results | $R^2$ | Test Results | $R^2$ |
| | 0.9825 | H0 accepted | 0.9815 | H0 accepted | 0.9817 | 0.9870 | H0 accepted | 0.9875 | H0 accepted | 0.9882 |
| AREQ | 0.9835 | H0 accepted | 0.9808 | H0 accepted | | 0.9883 | H0 accepted | 0.9888 | H0 accepted | |
| | 0.9816 | H0 accepted | | | | 0.9888 | H0 accepted | | | |
| | 0.9935 | H0 accepted | 0.9945 | H0 accepted | 0.9944 | 0.9897 | H0 accepted | 0.9920 | H0 accepted | 0.9944 |
| CRO1 | 0.9892 | **H0 rejected** | 0.9925 | **H0 accepted** | | 0.9940 | H0 accepted | 0.9808 | **H0 rejected** | |
| | 0.9930 | H0 accepted | | | | 0.9909 | H0 accepted | | | |
| | 0.8492 | H0 accepted | 0.9085 | H0 accepted | 0.9065 | 0.9745 | H0 accepted | 0.9776 | H0 accepted | 0.9801 |
| DAV1 | 0.9139 | **H0 rejected** | 0.9241 | H0 accepted | | 0.9746 | H0 accepted | 0.9619 | **H0 rejected** | |
| | 0.8434 | **H0 rejected** | | | | 0.9521 | **H0 rejected** | | | |
| | 0.9902 | **H0 rejected** | 0.9892 | H0 accepted | 0.9916 | 0.9916 | H0 accepted | 0.9912 | H0 accepted | 0.9941 |
| GUAM | 0.9872 | H0 accepted | 0.9885 | H0 accepted | | 0.9906 | H0 accepted | 0.9942 | H0 accepted | |
| | 0.9877 | **H0 rejected** | | | | 0.9931 | **H0 rejected** | | | |
| | 0.9985 | **H0 rejected** | 0.9989 | **H0 rejected** | 0.9995 | 0.9971 | **H0 rejected** | 0.9983 | H0 accepted | 0.9988 |
| HOB2 | 0.9984 | **H0 rejected** | 0.9992 | H0 accepted | | 0.9985 | H0 accepted | 0.9985 | H0 accepted | |
| | 0.9967 | **H0 rejected** | | | | 0.9978 | **H0 rejected** | | | |
| | 0.9161 | **H0 rejected** | 0.9546 | H0 accepted | 0.9521 | 0.9695 | H0 accepted | 0.9684 | H0 accepted | 0.9702 |
| KERG | 0.9618 | **H0 rejected** | 0.9464 | H0 accepted | | 0.9669 | H0 accepted | 0.9683 | H0 accepted | |
| | 0.9381 | H0 accepted | | | | 0.9607 | **H0 rejected** | | | |
| | 0.9994 | **H0 rejected** | 0.9995 | **H0 rejected** | 0.9995 | 0.9987 | H0 accepted | 0.9989 | H0 accepted | 0.9990 |
| KIRU | 0.9994 | **H0 rejected** | 0.9994 | H0 accepted | | 0.9989 | H0 accepted | 0.9989 | H0 accepted | |
| | 0.9987 | **H0 rejected** | | | | 0.9988 | H0 accepted | | | |
| | 0.9997 | **H0 rejected** | 0.9998 | H0 accepted | 0.9999 | 0.9997 | H0 accepted | 0.9997 | H0 accepted | 0.9998 |
| MATE | 0.9995 | **H0 rejected** | 0.9997 | **H0 rejected** | | 0.9997 | H0 accepted | 0.9997 | H0 accepted | |
| | 0.9998 | **H0 rejected** | | | | 0.9995 | **H0 rejected** | | | |
| | 0.9989 | **H0 rejected** | 0.9993 | H0 accepted | 0.9993 | 0.9988 | H0 accepted | 0.9989 | H0 accepted | 0.9990 |
| NYAL | 0.9987 | **H0 rejected** | 0.9996 | H0 accepted | | 0.9982 | **H0 rejected** | 0.9988 | H0 accepted | |
| | 0.9988 | H0 accepted | | | | 0.9976 | **H0 rejected** | | | |
| | 0.9987 | **H0 rejected** | 0.9986 | **H0 rejected** | 0.9993 | 0.9991 | H0 accepted | 0.9993 | H0 accepted | 0.9993 |
| POL2 | 0.9993 | H0 accepted | 0.9992 | H0 accepted | | 0.9994 | H0 accepted | 0.9993 | H0 accepted | |
| | 0.9993 | H0 accepted | | | | 0.9990 | H0 accepted | | | |
| | 0.9965 | **H0 rejected** | 0.9977 | H0 accepted | 0.9974 | 0.9960 | H0 accepted | 0.9974 | H0 accepted | 0.9974 |
| REYK | 0.9962 | H0 accepted | 0.9964 | H0 accepted | | 0.9976 | H0 accepted | 0.9964 | H0 accepted | |
| | 0.9946 | **H0 rejected** | | | | 0.9962 | H0 accepted | | | |
| | 0.9994 | H0 accepted | 0.9995 | **H0 rejected** | 0.9996 | 0.9996 | H0 accepted | 0.9996 | H0 accepted | 0.9997 |
| TOW2 | 0.9994 | H0 accepted | 0.9997 | H0 accepted | | 0.9996 | H0 accepted | 0.9997 | H0 accepted | |
| | 0.9989 | **H0 rejected** | | | | 0.9996 | H0 accepted | | | |
| | 0.9996 | H0 accepted | 0.9996 | H0 accepted | 0.9995 | 0.9993 | **H0 rejected** | 0.9995 | H0 accepted | 0.9997 |
| TRAK | 0.9994 | H0 accepted | 0.9995 | H0 accepted | | 0.9993 | **H0 rejected** | 0.9997 | H0 accepted | |
| | 0.9992 | H0 accepted | | | | 0.9978 | **H0 rejected** | | | |
| | 0.9811 | H0 accepted | 0.9810 | H0 accepted | 0.9814 | 0.9790 | H0 accepted | 0.9824 | H0 accepted | 0.9826 |
| USUD | 0.9836 | H0 accepted | 0.9826 | H0 accepted | | 0.9826 | H0 accepted | 0.9826 | H0 accepted | |
| | 0.9827 | H0 accepted | | | | 0.9796 | H0 accepted | | | |
| | 0.9996 | H0 accepted | 0.9997 | H0 accepted | 0.9997 | 0.9995 | H0 accepted | 0.9996 | H0 accepted | 0.9996 |
| VILL | 0.9994 | **H0 rejected** | 0.9996 | **H0 rejected** | | 0.9993 | **H0 rejected** | 0.9996 | H0 accepted | |
| | 0.9996 | H0 accepted | | | | 0.9995 | H0 accepted | | | |
| | 0.9995 | **H0 rejected** | 0.9996 | **H0 rejected** | 0.9998 | 0.9993 | **H0 rejected** | 0.9995 | H0 accepted | 0.9997 |
| WTZR | 0.9997 | **H0 rejected** | 0.9998 | H0 accepted | | 0.9996 | H0 accepted | 0.9998 | H0 accepted | |
| | 0.9998 | H0 accepted | | | | 0.9997 | H0 accepted | | | |
| | 0.9970 | **H0 rejected** | 0.9985 | **H0 rejected** | 0.9965 | 0.9979 | H0 accepted | 0.9983 | H0 accepted | 0.9986 |
| YELL | 0.9975 | H0 accepted | 0.9933 | **H0 rejected** | | 0.9987 | H0 accepted | 0.9966 | **H0 rejected** | |
| | 0.9891 | **H0 rejected** | | | | 0.9897 | **H0 rejected** | | | |
| | 0.9998 | **H0 rejected** | 0.9999 | **H0 rejected** | 0.9998 | 0.9997 | H0 accepted | 0.9997 | H0 accepted | 0.9997 |
| ZIMM | 0.9994 | **H0 rejected** | 0.9995 | **H0 rejected** | | 0.9991 | **H0 rejected** | 0.9994 | **H0 rejected** | |
| | 0.9998 | H0 accepted | | | | 0.9996 | H0 accepted | | | |

**Table 5.** For the up component, $R^2$ values and hypothesis test results for relative positioning and PPP.

| Stations | GAMIT/GLOBK v10.60 | | | | | GIPSY/OASIS II v6.3 | | | | |
|---|---|---|---|---|---|---|---|---|---|---|
| | 8 h | | 12 h | | 24 h | 8 h | | 12 h | | 24 h |
| | $R^2$ | Test Results | $R^2$ | Test Results | $R^2$ | $R^2$ | Test Results | $R^2$ | Test Results | $R^2$ |
| AREQ | 0.2631 | **H0 rejected** | 0.2216 | **H0 rejected** | 0.3542 | 0.4916 | **H0 rejected** | 0.5595 | H0 accepted | 0.5372 |
| | 0.2445 | **H0 rejected** | 0.1730 | **H0 rejected** | | 0.4303 | **H0 rejected** | 0.4172 | H0 rejected | |
| | 0.1148 | **H0 rejected** | | | | 0.3689 | **H0 rejected** | | | |
| CRO1 | 0.3195 | **H0 rejected** | 0.3316 | **H0 rejected** | 0.4295 | 0.2017 | **H0 rejected** | 0.2893 | **H0 rejected** | 0.4269 |
| | 0.3564 | **H0 rejected** | 0.2437 | **H0 rejected** | | 0.2525 | **H0 rejected** | 0.2026 | **H0 rejected** | |
| | 0.2114 | **H0 rejected** | | | | 0.2185 | **H0 rejected** | | | |
| DAV1 | 0.2535 | **H0 rejected** | 0.2231 | **H0 rejected** | 0.2382 | 0.2246 | **H0 rejected** | 0.3060 | **H0 rejected** | 0.2259 |
| | 0.1533 | **H0 rejected** | 0.1747 | **H0 rejected** | | 0.2042 | **H0 rejected** | 0.0599 | **H0 rejected** | |
| | 0.1072 | **H0 rejected** | | | | 0.0528 | **H0 rejected** | | | |
| GUAM | 0.1177 | **H0 rejected** | 0.1343 | **H0 rejected** | 0.1125 | 0.0513 | **H0 rejected** | 0.0490 | **H0 rejected** | 0.0571 |
| | 0.0462 | **H0 rejected** | 0.0999 | **H0 rejected** | | 0.0184 | **H0 rejected** | 0.0469 | **H0 rejected** | |
| | 0.1015 | **H0 rejected** | | | | 0.0427 | **H0 rejected** | | | |
| HOB2 | 0.0387 | **H0 rejected** | 0.0282 | **H0 rejected** | 0.0546 | 0.1889 | **H0 rejected** | 0.2199 | **H0 rejected** | 0.2894 |
| | 0.0615 | **H0 rejected** | 0.0620 | **H0 rejected** | | 0.1836 | **H0 rejected** | 0.1611 | **H0 rejected** | |
| | 0.0567 | **H0 rejected** | | | | 0.0958 | **H0 rejected** | | | |
| KERG | 0.2870 | **H0 rejected** | 0.3320 | **H0 rejected** | 0.4723 | 0.0932 | **H0 rejected** | 0.1247 | **H0 rejected** | 0.1993 |
| | 0.3759 | **H0 rejected** | 0.4180 | **H0 rejected** | | 0.0992 | **H0 rejected** | 0.1478 | **H0 rejected** | |
| | 0.3579 | **H0 rejected** | | | | 0.1337 | **H0 rejected** | | | |
| KIRU | 0.8169 | H0 accepted | 0.8241 | H0 accepted | 0.8174 | 0.8280 | H0 accepted | 0.8331 | H0 accepted | 0.8425 |
| | 0.7776 | H0 accepted | 0.7961 | H0 accepted | | 0.8169 | **H0 rejected** | 0.8185 | H0 accepted | |
| | 0.7893 | H0 accepted | | | | 0.8184 | H0 accepted | | | |
| MATE | 0.2250 | **H0 rejected** | 0.1726 | **H0 rejected** | 0.2347 | 0.0930 | **H0 rejected** | 0.1125 | **H0 rejected** | 0.2044 |
| | 0.1861 | **H0 rejected** | 0.2603 | **H0 rejected** | | 0.1218 | **H0 rejected** | 0.1824 | **H0 rejected** | |
| | 0.1653 | **H0 rejected** | | | | 0.1254 | **H0 rejected** | | | |
| NYAL | 0.9689 | **H0 rejected** | 0.9691 | **H0 rejected** | 0.9783 | 0.9628 | **H0 rejected** | 0.9646 | **H0 rejected** | 0.9733 |
| | 0.9547 | **H0 rejected** | 0.9673 | **H0 rejected** | | 0.9518 | **H0 rejected** | 0.9635 | **H0 rejected** | |
| | 0.9612 | **H0 rejected** | | | | 0.9584 | **H0 rejected** | | | |
| POL2 | 0.3080 | **H0 rejected** | 0.2624 | **H0 rejected** | 0.3324 | 0.2000 | **H0 rejected** | 0.1977 | **H0 rejected** | 0.2095 |
| | 0.1290 | **H0 rejected** | 0.3037 | **H0 rejected** | | 0.1167 | **H0 rejected** | 0.1575 | **H0 rejected** | |
| | 0.2513 | **H0 rejected** | | | | 0.1790 | **H0 rejected** | | | |
| REYK | 0.2590 | **H0 rejected** | 0.2774 | H0 accepted | 0.2492 | 0.2705 | **H0 rejected** | 0.3187 | H0 accepted | 0.3112 |
| | 0.2942 | **H0 rejected** | 0.2525 | **H0 rejected** | | 0.3359 | **H0 rejected** | 0.2428 | **H0 rejected** | |
| | 0.2324 | **H0 rejected** | | | | 0.1716 | **H0 rejected** | | | |
| TOW2 | 0.3345 | **H0 rejected** | 0.2440 | **H0 rejected** | 0.3889 | 0.5033 | **H0 rejected** | 0.5034 | **H0 rejected** | 0.5657 |
| | 0.1781 | **H0 rejected** | 0.3598 | **H0 rejected** | | 0.3938 | **H0 rejected** | 0.4722 | **H0 rejected** | |
| | 0.3405 | **H0 rejected** | | | | 0.4446 | **H0 rejected** | | | |
| TRAK | 0.2662 | **H0 rejected** | 0.2262 | **H0 rejected** | 0.2887 | 0.2370 | **H0 rejected** | 0.3055 | **H0 rejected** | 0.3646 |
| | 0.0784 | **H0 rejected** | 0.2334 | **H0 rejected** | | 0.3616 | **H0 rejected** | 0.2880 | **H0 rejected** | |
| | 0.3202 | **H0 rejected** | | | | 0.2040 | **H0 rejected** | | | |
| USUD | 0.3990 | **H0 rejected** | 0.4883 | **H0 rejected** | 0.5123 | 0.4952 | **H0 rejected** | 0.5507 | **H0 rejected** | 0.5685 |
| | 0.4516 | **H0 rejected** | 0.5359 | H0 accepted | | 0.5178 | **H0 rejected** | 0.4920 | H0 accepted | |
| | 0.5662 | H0 accepted | | | | 0.4842 | **H0 rejected** | | | |
| VILL | 0.1156 | **H0 rejected** | 0.1620 | **H0 rejected** | 0.2615 | 0.3436 | **H0 rejected** | 0.4192 | **H0 rejected** | 0.4938 |
| | 0.1169 | **H0 rejected** | 0.2393 | **H0 rejected** | | 0.3105 | **H0 rejected** | 0.3591 | **H0 rejected** | |
| | 0.2488 | **H0 rejected** | | | | 0.2690 | **H0 rejected** | | | |
| WTZR | 0.3145 | **H0 rejected** | 0.4220 | **H0 rejected** | 0.4262 | 0.2357 | **H0 rejected** | 0.3047 | **H0 rejected** | 0.3060 |
| | 0.3689 | **H0 rejected** | 0.3815 | **H0 rejected** | | 0.2412 | **H0 rejected** | 0.2306 | **H0 rejected** | |
| | 0.3281 | **H0 rejected** | | | | 0.2087 | **H0 rejected** | | | |
| YELL | 0.9095 | **H0 rejected** | 0.9272 | H0 accepted | 0.9362 | 0.9287 | **H0 rejected** | 0.9394 | **H0 rejected** | 0.9494 |
| | 0.9106 | **H0 rejected** | 0.9274 | H0 accepted | | 0.9340 | **H0 rejected** | 0.9449 | H0 accepted | |
| | 0.9090 | **H0 rejected** | | | | 0.9369 | **H0 rejected** | | | |
| ZIMM | 0.4338 | **H0 rejected** | 0.4099 | **H0 rejected** | 0.4714 | 0.4233 | **H0 rejected** | 0.4324 | **H0 rejected** | 0.4900 |
| | 0.3095 | **H0 rejected** | 0.3615 | **H0 rejected** | | 0.3174 | **H0 rejected** | 0.4005 | **H0 rejected** | |
| | 0.2704 | **H0 rejected** | | | | 0.3433 | **H0 rejected** | | | |