# Peer review of "Assessment of geodetic velocities using GPS campaign measurements over long baseline lengths"

_Natural Hazards and Earth System Sciences, 2018_

## Referee Comment (RC1) · Anonymous Referee #1 · 5 Nov 2018

Based on these materials and my own reading of the manuscript, I find that the content of your manuscript may merit publication in Natural Hazards and Earth System Sciences (NHESS) after a major revision.

---

## Referee Comment (RC2) · Anonymous Referee #2 · 8 Nov 2018

Review of "Accuracy of geodetic site velocities from repeated GPS measurements: relative positioning over long baselines" by Duman and Sanli.

In this paper, the authors performed and tested relative positioning at cGPS stations in an effort to assess the accuracy of the geodetic site velocities when these are derived by means of episodic measurements, i.e. as in GPS campaigns. Their analysis was applied at IGS stations and the obtained results were additionally compared with PPP solutions. The topic of this study is of interest for the "Natural Hazards and Earth System Science" readership. The manuscript is slightly novel and accordingly in order to warrant publication, the authors should address a number of important comments

that lead to "major" revisions, although in fact my suggestions are straightforward. My comments are reported in the following and I hope these may be of help to the authors for their revision work.

1) The authors need to rework the abstract. The problem with the abstract is that it reads too much like an introduction. Abstracts should concisely say what the authors did and what they found, so I suggest to rewrite it keeping in mind that abstract doesn't need to be verbose.

2) In the Introduction the authors should try to highlight the added value and the novelties of the present paper. In this context, they need to state the problem better, i.e. why is this study needed, and they need to make a better case for their study in this section. The case can be made by first reviewing what has been done in other studies towards the comparison of the geodetic velocities derived from continuous and episodic measurements. Then, they should define what they want to improve with their study and at the end of the introduction how they achieved it. They should also add a few more references about studies where GPS velocity fields have been used to facilitate tectonic and geodynamic research (e.g. Vernant et al. 2004 Geophysical Journal International; Serpelloni et al. 2007 Geophysical Journal International; Chousianitis et al. 2015 Journal of Geophysical Research) and make a brief assessment of the uncertainties in the velocity fields of these studies in comparison to velocity fields derived only via episodic measurements.

3) The authors do not mention sufficient details about their processing scheme in Gamit/Globk. Accordingly, they should add info about this, since Gamit/Globk has numerous options and the potential readers should be aware of the critical choices that the authors made. Also, have they combined their loosely-constrained daily solutions with daily global solutions for the whole IGS network in their second processing step? Finally, they should add more details regarding the realization of the reference frame and the way they adjusted their velocity data in the ITRF. Have they implemented the frame realization through "generalized constraints", have they applied a few iterations

to eliminate bad sites and to compute station weights for the reference frame stabilization? What criteria they have used to characterize the set of IGS stations that they used as reliable? Please be more specific.

---

## Referee Comment (RC3) · Anonymous Referee #1 · 18 Nov 2018

Dear authors and editor

I will attach you the revised version of you manuscript including some suggestions and questions about the GNSS data processing. In my point of view, I believe that article needs some improvements, which are related with the Glossary, Grammatical and some about the Graphical representation.

Please also note the supplement to this comment: https://www.nat-hazards-earth-syst-sci-discuss.net/nhess-2018-258/nhess-2018-258-RC3-supplement.pdf

[Figure]

[Figure]

**Supplement:**

[revised manuscript text omitted]

---

## Author Comment (AC1) · 23 Jan 2019

**Title**

**Reviewer**
recommended the title as being "Assessment of geodetic velocities using campaign measurements over long baseline length"

**Authors**
The title will be changed to "Assessment of geodetic velocities using campaign measurements over long baseline lengths" with a little modification to the last word "length" as suggested by the reviewer.

**Abstract**

**Reviewer**
Line 1, Remove "Currently" and "(i.e. repeated GPS measurements)" from the text

**Authors**
The abstract has now been rewritten considering the recommendations from Reviewer 2. Therefore only "Currently" and "(i.e. repeated GPS measurement)" have been removed from line 1 and the rest of the abstract has been recompiled (see the supplement file attached).

**Introduction**

**Reviewer**
Replace "The coordinates of a new point installed in a study area are usually found either by using relative point positioning or precise point positioning (PPP)" with "The coordinates of a new established site (benchmark or GNSS station) are usually found either by using relative point positioning or precise point positioning (PPP).

**Authors**
The introduction, as with the recommendation from Reviewer 2, has been rewritten. While recompiling the introduction, as recommended by the reviewer we used the term "campaign" but not "repeated" and added the reference Bitharis et. al., 2016 and Hollenstein et. al., 2008 (see the supplement file attached).

**GPS data analysis**

**Reviewer**
Please give me more details about the GPS data analysis strategy. For example:
1) which Mapping function you use?
2) A very important issue in relative positioning is the resolving ambiguities, can you provide these values/percentage of Wide lane and Narrow lane?
3) Which OTL model you choose?

**Authors**
1) We used Global Pressure and Temperature 2 (GPT2) (Lagler et. al., 2013) for both GAMIT and GIPSY processing. We modified the text accordingly.
2) Wide and Narrow lane phase ambiguities are provided for GAMIT processing with Figure 1 attached.

For GIPSY processing, unfortunately we only kept stacov (i.e. the file containing only position information and correlations between coordinates) files.

3) FES2004 ocean tide loading model from OSO Chalmer and the ocean tide model developed by Desai (2002) were used for the processing of GAMIT/GLOBK and GIPSY/OASIS II respectively.

**Page 3, Lines 4-6**

**Reviewer**

These is a very sensitive step, You use GLRED for Time-series daily reliabilities or

GLOBK? Also, you choose to estimate both EOPs? or you use weighted constrained, may have these values? Please take care on this step, because the recommended method/parameters dependents on the GNSS network scale.

**Authors**
GLOBK module was used for the combined solution. We chose to estimate the IERS (International Earth Rotation and Reference Systems Service) Bulletin B values for Earth rotation. The text has now been modified accordingly. Initially 18 IGS stations were selected for the realization of the reference frame. A reliable set of the stations was determined applying 4 iterations. Bad stations were eliminated and about 12 through 14 IGS stations were reliably used. This information is now included in the text (see the supplement file).

**Reviewer**
Can you provide some values of the daily Transformation residuals, a simpe a?

**Authors**
Daily transformation residuals for all sub-sessions listed in Table 1 are 0.095, 0.059, and 0.055 mas (milliarcseconds) for X, Y, and Z rotations as well as -15, 8, and -12 mm for X, Y, and Z translations. Given values are the mean of the translations for all days from 2000 to 2015.

Please also note the supplement to this comment:
https://www.nat-hazards-earth-syst-sci-discuss.net/nhess-2018-258/nhess-2018-258-AC1-supplement.pdf

[Figure]

**Fig. 1.** Wide and narrow lane fixed phase ambiguities in percentage

**Supplement:**

**ASSESSMENT OF GEODETIC VELOCITIES USING CAMPAIGN MEASUREMENTS OVER LONG BASELINE LENGTH**

[revised manuscript text omitted]

Trần, Đ. T., Nguyễn, T. Y., Dương, C. C., Vy, Q. H., Zuchiewicz, W., and Nguyễn, V. N.: Recent
crustal movements of northern Vietnam from GPS data. Journal of Geodynamics, 69, 5-
10, 2013.

Vernant, P., Nilforoushan, F., Hatzfeld, D., Abbassi, M. R., Vigny, C., Masson, F., ... and Tavakoli,
F.: Present-day crustal deformation and plate kinematics in the Middle East constrained by
GPS measurements in Iran and northern Oman. Geophysical Journal International, 157(1),
381-398, 2004.

Zhang, J., Bock, Y., Johnson, H., Fang, P., Williams, S., Genrich, J., ... and Behr, J.: Southern
California Permanent GPS Geodetic Array: Error analysis of daily position estimates and
site velocities. Journal of geophysical research: solid earth, 102(B8), 18035-18055, 1997.

Zumberge, J., Heflin, M., Jefferson, D., Watkins, M., and Webb, F. H.: Precise point positioning for the efficient and robust analysis of GPS data from large networks, Journal of

Geophysical Research: Solid Earth, 102, 5005-5017, 1997.

---

## Author Comment (AC2) · 23 Jan 2019

**Reviewer**

1) The authors need to rework the abstract. The problem with the abstract is that it reads too much like an introduction. Abstracts should concisely say what the authors did and what they found, so I suggest to rewrite it keeping in mind that abstract doesn't need to be verbose.

**Authors**

The Abstract has now been shortened excluding unnecessary statements (see the

supplement file).

**Reviewer**
2) (a) In the Introduction the authors should try to highlight the added value and the novelties of the present paper. In this context, they need to state the problem better, i.e. why is this study needed, and they need to make a better case for their study in this section. The case can be made by first reviewing what has been done in other studies towards the comparison of the geodetic velocities derived from continuous and episodic measurements.
(b) Then, they should define what they want to improve with their study and at the end of the introduction how they achieved it.
(c) They should also add a few more references about studies where GPS velocity fields have been used to facilitate tectonic and geodynamic research (e.g. Vernant et al. 2004 Geophysical Journal International; Serpelloni et al. 2007 Geophysical Journal International; Chousianitis et al. 2015 Journal of Geophysical Research) and make a brief assessment of the uncertainties in the velocity fields of these studies in comparison to velocity fields derived only via episodic measurements.

**Authors**
The introduction has now been recompiled taking into account the above suggestions. The above stated literature was also included in the new introduction (see the supplement file).

**Reviewer**
3) (a) The authors do not mention sufficient details about their processing scheme in Gamit/Globk. Accordingly, they should add info about this, since Gamit/Globk has

numerous options and the potential readers should be aware of the critical choices that the authors made.

**Authors**
With also recommendation by Reviewer 1 we now include some details about our GAMIT/GLOBK processing. The text also has now been modified accordingly (see the supplement file).

**Reviewer**
(b) Also, have they combined their loosely-constrained daily solutions with daily global solutions for the whole IGS network in their second processing step?

**Authors**
Since our network contained IGS stations from globally scattered stations we did not extra combine our loosely-constrained daily solutions with any other global solution.

**Reviewer**
(c) Finally, they should add more details regarding the realization of the reference frame and the way they adjusted their velocity data in the ITRF. Have they implemented the frame realization through "generalized constraints", have they applied a few iterations to eliminate bad sites and to compute station weights for the reference frame stabilization?

**Authors**
Yes, "generalized constraints" were applied in the analysis. We selected 18 globally

distributed IGS stations and applied four iterations to eliminate bad sites as well as computing station weights. Then 13 reliable stations were left to realize the reference frame. The text has now been modified accordingly (see the supplement file).

**Reviewer**
(d) What criteria they have used to characterize the set of IGS stations that they used as reliable? Please be more specific.

**Authors**
To characterize the set of our IGS stations we used GPS days in which ionospheric kappa index is smaller than 4 (as also stated in the manuscript body), IGS stations that are distributed globally, 3 consecutive days with common data for all stations, over 95

Please also note the supplement to this comment:
https://www.nat-hazards-earth-syst-sci-discuss.net/nhess-2018-258/nhess-2018-258-AC2-supplement.pdf

---

## Author Response (AR1)

Dear Editor,

We once more thank to the Editor and the two anonymous reviewers for their constructive comments. We are submitting the corrected and revised manuscript nhess-2018-258 considering the comments of the anonymous reviewers RC1 and RC2.

Apart from that, we also include corrections to some minor points that we noticed before the submission. These mainly constitute:

**1 Page 1, Title, I guess the abbreviation "GPS" was missing in the title and we added that.**

**2 Page 3, Line 21, We included Cetin et al. 2018 which was forgotten in the supplementary document.**

**3 Page 5, Line 13, In the equation, we removed tilda on top of x and replaced it with a hat which represents regression values and is understood much more easily by the reader. Then in Line 14 we included the necessary explanation about the new notation.**

**4 As also suggested by Reviewer 1 for the title, we changed some of the "baselines" to "baseline lengths" to prevent confusion between "GPS baseline components" and "baseline distance between geodetic points".**

**5 We expanded the acknowledgement including the name of a colleague who was really helpful in the interpretation of GAMIT results. In addition, we also thanked to the two anonymous reviewers.**

The other details were already submitted as responses to the comments of the two anonymous reviewers.

Kind regards,

D. Ugur Sanli